# Holistic Reasoning with Long-Context LMs: A Benchmark for Database Operations on Massive Textual Data

**Seiji Maekawa**[*]  **Hayate Iso**[*]   **Nikita Bhutani**
Megagon Labs
{seiji, hayate, nikita}@megagon.ai

## Abstract

The rapid increase in textual information means we need more efficient methods to sift through, organize, and understand it all. While retrieval-augmented generation (RAG) models excel in accessing information from large document collections, they struggle with complex tasks that require aggregation and reasoning over information spanning across multiple documents–what we call *holistic reasoning*. Long-context language models (LCLMs) have great potential for managing large-scale documents, but their holistic reasoning capabilities remain unclear. In this work, we introduce HoloBench, a novel framework a novel framework that brings database reasoning operations into text-based contexts, making it easier to systematically evaluate how LCLMs handle holistic reasoning across large documents. Our approach adjusts key factors such as context length, information density, distribution of information, and query complexity to evaluate LCLMs comprehensively.

Our experiments show that the amount of information in the context has a bigger influence on LCLM performance than the actual context length. Furthermore, the complexity of queries affects performance more than the amount of information, particularly for different types of queries. Interestingly, queries that involve finding maximum or minimum values are easier for LCLMs and are less affected by context length, even though they pose challenges for RAG systems. However, tasks requiring the aggregation of multiple pieces of information show a noticeable drop in accuracy as context length increases. Additionally, we find that while grouping relevant information generally improves performance, the optimal positioning varies across models. Our findings surface both the advancements and the ongoing challenges in achieving a holistic understanding of long contexts. These can guide future developments in LCLMs and set the stage for creating more robust language models for real-world applications.

  Code        https://github.com/megagonlabs/holobench
  Benchmark   https://hf.co/datasets/megagonlabs/holobench

## 1  Introduction

The explosive growth of textual data has highlighted the need for efficient ways to process, organize, and comprehend large document collections. While retrieval-augmented generation (RAG) models have made accessing information from these vast sources easier (Lewis et al., 2020; Maekawa et al., 2024), these models are limited by their reliance on localized context retrieval. This makes them less effective for complex tasks that require *holistic reasoning*—the ability to synthesize, aggregate, and reason over information that spans across multiple documents. For instance, when asked, "Which company employed the most people?", traditional retrieval models might provide individual statistics for each company but fail to offer a comprehensive comparison across all relevant entities. This reveals a fundamental limitation: RAG systems excel at retrieving specific information but struggle to aggregate and reason over broader cross-document contexts. This gap highlights the need for

---

[*]Equal contribution.

| Benchmark | Variable context len. | Variable info. amount | Variable info. position | Reasoning over 100+ info. pieces | Verifiable answers |
|---|---|---|---|---|---|
| Needle-in-a-haystack (Kamradt, 2023) | ✓ | ✗ | ✓ | ✗ | ✓ |
| Multi Needles-in-a-haystack (Kamradt, 2023) | ✓ | ✓ | ✓ | ✗ | ✓ |
| En.QA/En.MC (Zhang et al., 2024) | ✗ | ✗ | ✗ | ✗ | ✗ human-anno. |
| BABILONG (Kuratov et al., 2024) | ✓ | ✗ | ✓ | ✗ | ✓ |
| NOCHA (Karpinska et al., 2024) | ✗ | ✗ | ✗ | ✗ | ✗ human-anno. |
| FLENQA (Levy et al., 2024) | ✓ | ✗ | ✓ | ✗ | ✓ |
| HOLOBENCH (Ours) | ✓ | ✓ | ✓ | ✓ | ✓ |

Table 1: Characteristics of Long-Context LM Benchmarks. "human-anno." indicates that gold answers of the benchmarks are created by human annotators.

more advanced models capable of holistic reasoning, particularly in tasks requiring multi-document synthesis, comparative analysis, and contextual integration across extensive datasets.

Long-Context Language Models (LCLMs) (OpenAI, 2024a; Anthropic, 2024; Reid et al., 2024; Dubey et al., 2024), which can process hundreds of thousands of tokens, present a promising solution to this challenge. With their ability to handle long text inputs, they can retrieve, aggregate, and reason over large amounts of relevant information, enabling a more holistic approach to complex, multi-conditional queries. However, existing studies primarily evaluate LCLMs as a substitute for retrieval models that focus on localized contexts rather than evaluating their capability for holistic reasoning (Kamradt, 2023; Lee et al., 2024; Zhang et al., 2024; Hsieh et al., 2024), as summarized in Table 1. This gap persists partly due to the absence of suitable benchmarks that rigorously test models on tasks requiring reasoning across extensive document collections.

In this paper, we introduce the Holistic Reasoning Benchmark (HoloBench), a new framework specifically designed to evaluate LCLMs' ability to perform holistic reasoning over long contexts. HoloBench leverages database reasoning operations to systematically evaluate how well models can aggregate, compare, and draw conclusions from distributed information. By adapting existing text-to-SQL benchmarks (Yu et al., 2018; Katsogiannis-Meimarakis & Koutrika, 2023; Li et al., 2024a), HoloBench enables an automated and scalable evaluation process, eliminating the need for labor-intensive manual annotations.

A key innovation of HoloBench is its ability to control three critical factors that influence LCLM performance: (1) the length of the context and the amount of information contained within, (2) the position of relevant information within the context, and (3) the type and difficulty of queries. These factors allow for a more comprehensive assessment of LCLMs' holistic reasoning capabilities. Existing benchmarks have primarily explored only one or two of these dimensions in isolation (Zhang et al., 2024; Hsieh et al., 2024; Levy et al., 2024), making HoloBench the first framework to evaluate LCLMs across all three dimensions simultaneously.

Through detailed experiments, we uncovered several important findings: 1) The amount of relevant information in the context impacts model performance much more than the total length of the context itself. 2) The complexity of the questions plays a more important role in performance than the total amount of information being processed. For example, models easily handle questions that ask for maximum or minimum values, but they struggle when the task requires aggregating several pieces of information. 3) How information is positioned in the text also matters. Some models work better when relevant information is grouped together, while others can handle more scattered information.

Our findings highlight both the progress and the remaining challenges in enabling models to perform holistic reasoning over long contexts. Our work addresses a critical gap in evaluating long-context models and offers valuable insights into their capabilities and limitations when processing large-scale textual data.

## 2 RELATED WORK

**Neural Databases** Thorne et al. (2021) have proposed a neural database architecture designed to process database-like queries over natural language text. While their approach demonstrates scalability for small to medium-sized datasets, it is not sufficient for LCLMs, as the database size and complexity of queries are limited. Moreover, obtaining large-scale ground truth annotations for such tasks is prohibitively expensive and labor-intensive, posing a challenge for comprehensive

evaluation in this area. Trappolini et al. (2023) extend neural databases to multimodal data, including images and text. However, their work focuses on multimodal retrieval, not on evaluating LCLMs for long-context tasks, leaving a gap in addressing the challenges of long-context understanding.

**Long-context Language Model Benchmarks** Many recent studies (Kamradt, 2023; Lee et al., 2024; Hsieh et al., 2024; Kuratov et al., 2024; Karpinska et al., 2024; Laban et al., 2024; Levy et al., 2024; Zhang et al., 2024) have introduced benchmarks to evaluate LCLMs on tasks that require understanding and processing extended contexts, including "needle-in-a-haystack" tasks, which involve locating specific "needles with magic numbers" within extensive texts. While these benchmarks effectively test models' retention and coherence across large texts, they predominantly focus on localized tasks or summarization. Specifically, most examples require aggregating up to approximately ten information pieces, which does not fully assess models' abilities to handle more complex reasoning tasks over vast amounts of information in lengthy documents.

For instance, En.QA in $\infty$BENCH (Zhang et al., 2024) includes only a few questions that require information aggregation over more than ten pieces. However, the gold answers for these questions are manually annotated by the authors, making the creation process prohibitively costly and prone to annotation errors. In contrast, we aim to investigate how LCLMs synthesize and integrate vast amounts of information distributed across extended contexts in an automated and scalable manner. By designing a corpus generation framework that can systematically vary the complexity and distribution of information, we address the limitations of existing benchmarks and provide a more comprehensive evaluation of LCLMs' holistic reasoning capabilities.

**Retrieval-Augmented Generation** Retrieval-Augmented Generation (RAG) (Lewis et al., 2020) combines the strengths of retrieval-based methods and generative language models to enhance the generation of accurate and contextually relevant responses. Recent studies (Li et al., 2024b; Yu et al., 2024) have discussed the pros and cons of both LCLMs and RAG. However, their analysis relies on the abovementioned benchmarks that are designed for localized tasks. The insights are not applicable to our task, which requires filtering information based on given complex constraints and aggregating a larger amount of information.

# 3 THE HOLISTIC REASONING BENCHMARK

The Holistic Reasoning Benchmark (HoloBench) is an evaluation framework designed to assess the ability of LCLMs to perform holistic reasoning over massive textual data. HoloBench leverages database operations to create complex reasoning tasks that require models to aggregate and synthesize information distributed across extensive contexts.

## 3.1 DESIGN PRINCIPLES

The design of HoloBench is motivated by the need to evaluate LCLMs on holistic reasoning tasks that require comprehensive understanding and reasoning over large-scale document collections. To achieve this, HoloBench is built upon the following key principles:

- **Complex Reasoning Tasks:** The benchmark includes tasks that require LCLMs to perform operations such as aggregation, comparison, and inference over massive textual data.

- **Controlled Evaluation Factors:** The benchmark systematically varies critical factors influencing LCLM performance, including context length, information density, information distribution, and query complexity.

- **Automated and Scalable Evaluation:** The benchmark dynamically generates gold answers based on evaluation parameters, eliminating the need for labor-intensive manual annotations.

To adhere to these principles, HoloBench utilizes text-to-SQL benchmarks to construct holistic reasoning data. This approach ensures that the benchmark provides verifiable answers by executing SQL queries on databases while dynamically controlling context size and information distribution.

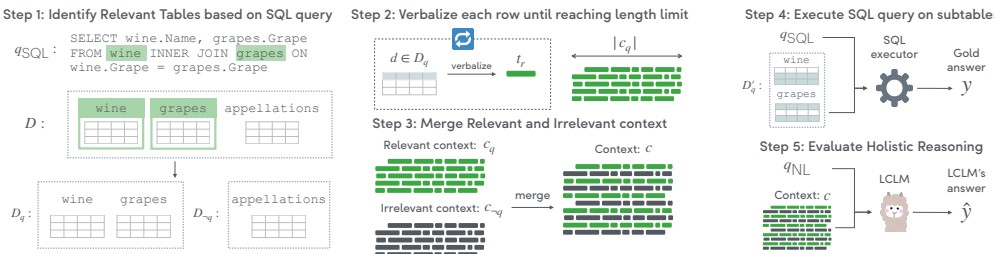

Figure 1: Overview of the HoloBench instance generation process, illustrating the partitioning of databases into relevant and non-relevant subsets, verbalization of table rows into textual contexts, and construction of the final inference context based on information positioning parameters.

## 3.2 BENCHMARK CREATION

HoloBench is synthesized from text-to-SQL benchmarks by adapting queries and their corresponding databases into natural language tasks. Each question $q_{\text{NL}}$ in HoloBench is paired with a SQL query $q_{\text{SQL}}$ and database $D$, which allows automatic generation of dynamic gold answers $y$ as evaluation parameters are varied. The benchmark creation process, illustrated in Figure 1, involves the following steps:

1. **Identifying Relevant Tables:** Based on the SQL query $q_{\text{SQL}}$, tables relevant to the query in the database $D = \{d_1, \ldots, d_n\}$ are identified and isolated into subset $D_q$, while the remainder are designated as $D_{\neg q}$. This process includes writing a pair of natural language questions $q_{\text{NL}}$ and its corresponding SQL query that can be executed on the database.

2. **Verbalizing Data Rows:** Each row $r$ from both the relevant tables $d \in D_q$ and the non-relevant tables $d \in D_{\neg q}$ is iteratively sampled and transformed into its natural language text $t_r$ using predefined templates. This verbalization continues until the lengths $|c_q|$ and $|c_{\neg q}|$ match the predetermined context lengths, ensuring a balanced representation of relevant and irrelevant data in line with the total input length $|c|$ and predefined information density.

3. **Merging Contexts:** The relevant context $c_q$ and the irrelevant context $c_{\neg q}$ are then merged based on an information positioning parameter that dictates the structure and placement of relevant and non-relevant information within the overall context $c$. This ensures that context $c$ is representative of both necessary and supplementary data, optimizing the distribution for evaluation.

4. **Executing SQL Queries:** An SQL executor runs $q_{\text{SQL}}$ on the subtables derived from $D_q'$ to extract the gold answer $y$. This answer is used to assess the accuracy of the LCLM's response $\hat{y}$ when provided with the context $c$ and the natural language question $q_{\text{NL}}$. The evaluation measures how effectively the model can utilize the structured context to generate correct responses.

This process enables the creation of a dynamic benchmark for holistic reasoning with desirable properties while requiring minimal to no human effort.

## 3.3 IMPLEMENTATION DETAILS

HoloBench is implemented using a variety of domains from the Spider dataset (Yu et al., 2018), i.e., `wine_1`, `college_2`, `flight_4`, `soccer_1`, and `store_1`.

We categorize query types into five groups based on SQL components:

1. **Aggregation** (e.g., SUM, AVG, and COUNT) which requires extracting relevant information and correctly aggregating it.
2. **Max/Min** (e.g., MAX, MIN), which involves finding extreme values within the data.
3. **Join** (e.g., JOIN and INNER JOIN), which requires two or more documents to retrieve the information requested in the query.
4. **Comparison** (e.g., >, <, =, and BETWEEN), which requires numeric comparisons to filter data
5. **Ranking** (ORDER BY and LIMIT) which requires identifying top-$k$ results and sorting them.

| Query Type | Natural language / SQL queries |
|---|---|
| Aggregation | What is the total number of unique instructors?
`SELECT` `count(DISTINCT instructor_name)` `FROM advisor;` |
| Max/Min | What is the maximum capacity of any classroom?
`SELECT` `MAX(capacity)` `FROM classroom;` |
| Join | List the course titles and the corresponding buildings.
`SELECT course.title, section.building FROM course`
`INNER JOIN section ON course.title = section.course_title;` |
| Comparison | Find the buildings which have rooms with capacity more than 50.
`SELECT DISTINCT building FROM classroom WHERE` `capacity > 50;` |
| Ranking | What are the titles of the top 5 posts with the highest popularity?
`SELECT Title FROM posts` `ORDER BY ViewCount DESC LIMIT 5;` |

Table 2: Categorization of query types with example natural language and SQL queries.

| Difficulty Level | Natural Language / SQL Queries |
|---|---|
| Easy | What is the name of the wine with the highest price?
`SELECT wine_name FROM wine WHERE price = (SELECT MAX(price) FROM wine);` |
| Medium | List the names of flights that depart from 'JFK' and arrive at 'LAX'.
`SELECT flight_name FROM flights WHERE departure_airport = 'JFK' AND arrival_airport = 'LAX';` |
| Hard | What are the names of colleges that have more than 10,000 students and are located in California?
`SELECT college_name FROM college WHERE num_students > 10000 AND state = 'California';` |

Table 3: Categorization of Query Difficulty Levels with Example Natural Language and Corresponding SQL Queries.

Figure 2: Illustration of how relevant context $c_q$ (highlighted in green) is placed within the entire context $c$ for different information positioning strategies.

Table 2 shows example queries for this categorization. Note that a single query can be associated with multiple types. These query types are automatically decided based on the type of database operator(s) in the SQL query. We also define three levels of query difficulty: **Easy**, **Medium**, and **Hard**. These are automatically decided based on the number of SQL components. Table 3 shows example queries by difficulty level. We provide more detailed definitions in Appendix A.3.1 and A.3.2. To observe the impact of information positioning, we consider three clustered positioning strategies –*Beginning*, *Middle* and *End*– and two distributed positioning strategies –*Uniform* and *Bimodal* (concentrated at both ends). These are illustrated in Figure 2.

For each database, the authors write six questions per query difficulty to cover all query types, resulting in 90 questions for HOLOBENCH since queries in the Spider dataset do not cover various query types. To ensure quality and variety, more than two authors reviewed the questions. We provide several examples in Appendix A.3.3, and the complete query set is available in the supplementary material. To convert each table row into a textual representation, we need to ensure that all necessary table information is included in the output while any extraneous details that could conflict with the ground truth answer $y$ is excluded. To achieve this, we develop a template-based text generation system. We manually analyze the queries in the seed dataset for each database to identify the relevant table columns needed for verbalization. Based on this analysis, we create custom templates for

each table that incorporate all required columns. Additionally, to ensure diversity in the generated text, we design five distinct templates for each table. Example templates are given in Table 4.

| Table/Database | Template |
|---|---|
| customers/store_1 | In {city}, {country}, {full_name} can be reached at {phone} for assistance from {support_rep_employee_name}. |
| airports/flight_4 | The {name} is located in {city}, {country} at an elevation of {elevation} feet. |
| Player/soccer_1 | With a height of {height} cm, {player_name} was born on {birthday} and weighs {weight} lb. |

Table 4: Templates for text generation. {placeholder} indicate column names that can be replaced with actual values from the tables.

## 4 EXPERIMENTS

### 4.1 SETUP

**Models:** We use nine LCLMs that support $128k$ or longer context length: Llama 3.1 8B/70B/405B-instruct (Dubey et al., 2024), GPT-4o-mini/GPT-4o (OpenAI, 2024a), o1-mini (OpenAI, 2024b), Claude 3.5 Sonnet (Anthropic, 2024), Gemini 1.5 Flash/Pro (Reid et al., 2024) and five more recent LCLMs: o3-mini (OpenAI, 2025), Gemini 2.0 Flash/Flash-Thinking, and DeepSeek-V3/R1 (DeepSeek-AI, 2024; 2025). We used the same prompt across all the models shown in Appendix A.4.

**Configurations:** We evaluate LCLM performance across varying context lengths and information amounts. The context lengths $|c|$ tested are 4k, 8k, 16k, 32k, and 64k tokens. To better understand how the amount and density of relevant information affect performance, we employ two setups: a) constant information amount: The total relevant information is fixed at 2k tokens, regardless of context length, and b) constant information density: The relevant information is distributed at a fixed density of 50% throughout the context.

**Evaluation Metrics:** We evaluated the holistic reasoning performance of LCLMs by determining whether each element in gold answer $y$ is included in the output $\hat{y}$ (Kamradt, 2023; Fu et al., 2024). Since the model produces the answer $\hat{y}$ in natural language, while the gold answer $y$ is a structured result set, we evaluate the performance on a per-row basis. We use the gpt-4o-mini to classify each row as *Exact Match*, *Partial Match*, or *No Match*, assigning scores of 100%, 50%, and 0%, respectively. This recall-oriented accuracy for each instance is calculated as the average score across all rows in the result set. We manually tested this evaluation metric for 20 questions, achieving a 93.8% agreement between the model evaluations and human judgments.[1]

We report three types of performance measures: the unweighted average (Avg.), and two weighted averages based on context length. (1) wAvg. (inc), where the weight increases linearly with context length, simulating scenarios where longer sequences dominate usage, and (2) wAvg. (dec), where the weight decreases linearly with context length. (Hsieh et al., 2024).

### 4.2 MAIN RESULTS

#### 4.2.1 EFFECT OF CONTEXT LENGTH AND INFORMATION AMOUNT

Table 5 summarizes the performance of various LCLMs as the average score across all questions in HoloBench. Specifically, we present performances for different context lengths under both constant information amount and constant information density settings. Note that we use a uniform positioning strategy for the experiments.

---

[1] In a preliminary study, we experimented with a basic 5-point scoring system (Liu et al., 2023; Padlewski et al., 2024), but it demonstrated low alignment with human judgments.

| LCLMs | Constant Information Amount (2k) | | | | | | | | Constant Information Density (50%) | | | | | | | |
|---|---|---|---|---|---|---|---|---|---|---|---|---|---|---|---|---|
| | 4k | 8k | 16k | 32k | 64k | Avg. | wAvg. (inc) | wAvg. (dec) | 4k | 8k | 16k | 32k | 64k | Avg. | wAvg. (inc) | wAvg. (dec) |
| Llama-3.1-405b | 64.0 | 61.0 | 57.5 | 55.1 | 44.8 | 56.5 | 53.5 | 59.4 | 64.0 | 59.4 | 55.1 | 41.4 | 30.2 | 50.0 | 44.3 | 55.7 |
| GPT-4o | 62.5 | 52.2 | 53.6 | 59.4 | 52.0 | 55.9 | 55.0 | 56.8 | 62.5 | 57.5 | 45.3 | 43.3 | 37.5 | 49.2 | 45.0 | 53.5 |
| o1-mini | 63.5 | 55.8 | 48.8 | 47.6 | 40.3 | 51.2 | 47.5 | 54.8 | 63.5 | 51.2 | 45.8 | 34.4 | 29.2 | 44.8 | 39.1 | 50.5 |
| Claude-3.5-Sonnet | 49.3 | 52.2 | 46.4 | 42.3 | 41.9 | 46.4 | 44.8 | 48.1 | 49.3 | 50.7 | 47.2 | 42.1 | 34.7 | 44.8 | 42.3 | 47.3 |
| Llama-3.1-70b | 60.5 | 51.2 | 41.1 | 42.2 | 33.6 | 45.7 | 41.5 | 49.9 | 60.5 | 53.7 | 45.9 | 39.0 | 28.6 | 45.5 | 40.3 | 50.8 |
| GPT-4o-mini | 51.5 | 50.1 | 43.1 | 43.2 | 38.7 | 45.3 | 43.2 | 47.5 | 51.5 | 45.6 | 37.8 | 35.0 | 26.3 | 39.2 | 35.2 | 43.3 |
| Gemini-1.5 Pro | 45.6 | 43.4 | 32.8 | 39.2 | 42.2 | 40.6 | 39.9 | 41.4 | 45.6 | 35.1 | 31.9 | 30.0 | 31.5 | 34.8 | 32.6 | 37.0 |
| Gemini-1.5 Flash | 36.8 | 32.0 | 34.3 | 27.2 | 26.8 | 31.4 | 29.8 | 33.1 | 36.8 | 30.7 | 25.8 | 27.7 | 20.5 | 28.3 | 25.9 | 30.7 |
| Llama-3.1-8b | 39.1 | 27.4 | 28.7 | 9.8 | 10.0 | 23.0 | 18.0 | 28.1 | 39.1 | 34.7 | 21.0 | 6.6 | 1.1 | 20.5 | 13.5 | 27.4 |
| Gemini-2.0 Flash | 66.2 | 62.3 | 59.1 | 57.3 | 53.0 | 59.6 | 57.5 | 61.7 | 66.2 | 60.1 | 57.0 | 52.5 | 48.2 | 56.8 | 53.9 | 59.7 |
| Gemini-2.0 Flash Thinking | 69.5 | 54.2 | 59.5 | 59.6 | 43.0 | 57.2 | 54.0 | 60.3 | 69.5 | 55.3 | 51.9 | 43.2 | 30.3 | 50.1 | 44.0 | 56.1 |
| o3-mini | 68.6 | 62.6 | 51.6 | 46.8 | 42.7 | 54.5 | 50.0 | 59.0 | 68.6 | 58.0 | 47.2 | 33.6 | 22.4 | 46.0 | 38.2 | 53.8 |
| DeepSeek-V3 | 59.5 | 51.7 | 50.8 | 47.3 | 53.6 | 52.6 | 51.5 | 53.7 | 59.5 | 56.9 | 52.5 | 45.8 | 36.0 | 50.1 | 46.2 | 54.0 |
| DeepSeek-R1 | 51.1 | 41.8 | 34.1 | 24.1 | 36.4 | 37.5 | 34.4 | 40.7 | 51.1 | 60.4 | 31.9 | 24.6 | 11.3 | 35.9 | 28.2 | 43.6 |

Table 5: Model performances for different context lengths under both constant information amount and constant information density settings. *Avg.* refers to the average across context sizes, and *wAvg.* denotes the weighted average, with weights either increasing (inc) or pydecreasing (dec) linearly with context length. The best score of each column is underlined. Generally, performance declines as context length increases. The drop is wider for information density than information amount.

**Information amount impacts more than context length** The experimental results indicate that the amount of information has a more significant impact on LCLM performance than context length. For instance, Llama-3.1-405B's accuracy drops from 64.0% to 30.2% as information density increases, compared to a smaller drop to 44.8% when only the context length increases. Similarly, GPT-4o achieves 52.0% accuracy with a 64k context and 2k of relevant information, but performance decreases to 45.3% when processing 8k of information in a shorter 16k context (with 50% information density). These results suggest that increased information density negatively impacts accuracy more than simply extending the context length.

**Llama-3.1-405b is best for short contexts, while GPT-4o dominates in long contexts** Overall, Llama-3.1-405b achieves the highest average and weighted average (dec) scores, indicating strong performance with shorter contexts. We observe this trend in both information amount and information density settings. However, GPT-4o achieves notably higher average (inc) score than Llama-3.1-405b (55.0 vs 53.5), indicating GPT-4o is more consistent and robust in handling longer contexts. This is also evident in the significantly higher scores achieved by GPT-4o for 32k and 64k context lengths.

**Model size is key for holistic reasoning in long contexts** The smaller models used in this study (GPT-4o-mini, Llama-3.1-8b, Gemini-1.5 Flash) claimed high accuracy in retrieval tasks such as the Needle-in-a-haystack test. However, for tasks that require holistic reasoning, model size significantly affects performance, especially as context length increases. For instance, in the Llama-3.1 series, the 8b model achieved only 1% accuracy under the 64k context with 50% information density, whereas the 405B model reached around 30% accuracy under the same conditions. This trend is consistently observed across other model series as well.

**Reasoning-focused LCLMs struggle with extended contexts** Reasoning-focused models—o1-mini, o3-mini, Gemini-2.0 Flash Thinking, and DeepSeek-R1—show strong performance on shorter contexts but experience significant performance degradation as context length increases. This suggests that current training approaches for reasoning-focused models may not adequately address the complexities of processing longer contexts. While general-purpose models like Gemini-2.0 Flash maintain consistent performance across context lengths, the challenges faced by reasoning-focused models highlight an important area for future research: developing training strategies that can maintain robust reasoning capabilities even with extended contexts.

### 4.2.2 Effect of Information Position

For brevity, we focus on the best-performing models, Llama-3.1-405b and GPT-4o, to examine the effect of information position. We use constant information density as the baseline setting unless

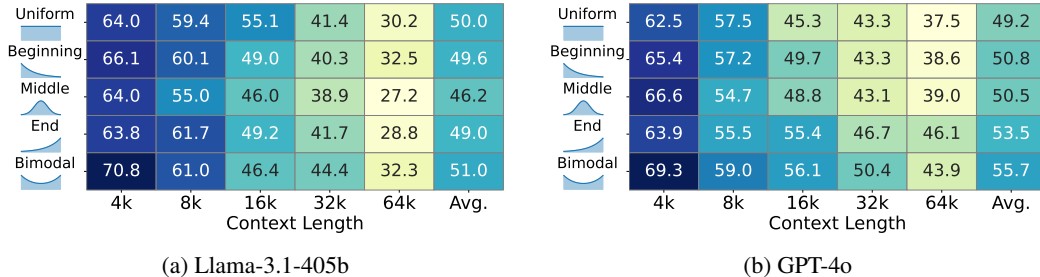

(a) Llama-3.1-405b            (b) GPT-4o

Figure 3: Variations in model performances with position of relevant information. Concentrated information benefits most models, although optimal position varies for each model.

otherwise specified. Figure 3 summarizes the model performances across five information positions: *uniform*, *beginning*, *middle*, *end*, and *bimodal*.

**Llama-3.1-405b is robust for information position, while GPT-4o is more sensitive**   Llama-3.1-405b maintains stable performance across various information positions, showing resilience whether the information is distributed uniformly or clustered. In contrast, GPT-4o performs better when relevant information is placed at the end or bimodal, but its accuracy declines when the information is grouped at the beginning and middle, suggesting that it tends to favor details presented at the end.

### 4.2.3 EFFECT OF QUERY TYPE AND DIFFICULTY

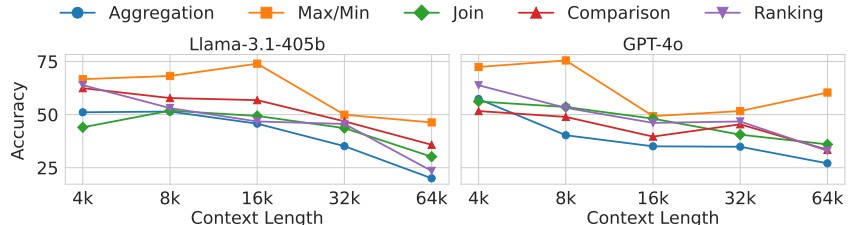

Figure 4: LCLMs' performance on various query types. Max/Min queries are an easier task across context lengths than other types. In contrast, We observe a performance drop for Aggregation queries, particularly for longer contexts.

Next, we evaluate the performance of Llama-3.1-405b and GPT-4o using constant information density and uniform information placement, considering query type and difficulty.

**Query type affects performance: Max/Min is easier while aggregation is harder.** Figure 4 summarizes the performance of models across different query types. As shown, model performance varies significantly depending on the query type. Max/Min queries are consistently easier than other types of queries. This is likely because extreme value retrieval tasks are relatively easier and do not scale in complexity with larger context lengths, unlike other tasks.

Interestingly, GPT-4o performs on par with other query types for Aggregation queries at 4k context length. However, as the context length increases, there is a notable drop in performance, indicating the model struggles to aggregate relevant information effectively in longer contexts. To investigate this behavior, we select a random aggregation query: *"Find the destination airport and its country that receives flights from the highest number of distinct source airports, and include that number in the result."* and inspect the model predictions. We find that GPT-4o correctly lists all relevant information and produces the correct answer when the context length is 4k. At 8k context length, it can find relevant information but begins to struggle with correctly retrieving and aggregating the information. Specifically, it returns the correct airports but miscounts the number of airports. At 64k context length, it skips the detailed analysis of the content and instead jumps straight to answering the question, leading to incorrect responses. We provide model outputs are in Figures 14, 15, and 16 in Appendix. This highlights the limitation of LCLMs' step-by-step reasoning abilities as the context size increases.

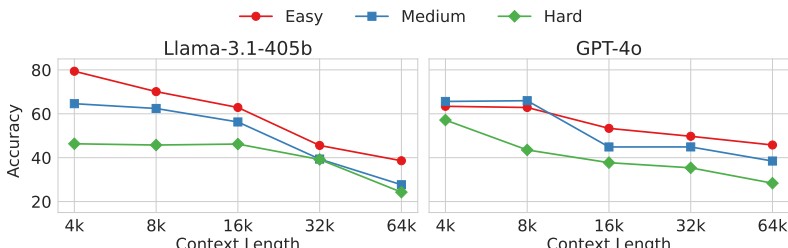

Figure 5: LCLMs' performance on various query difficulties. Model performance declines as query difficulty increases. Performance drops further as context length increases.

**Query complexity often affects performance more than the total volume of information.** Figure 5 summarizes the performance of Llama-3.1-405b and GPT-4o across different query difficulties. We observe that the relative order of difficulty—easy, medium, and hard—remains consistent across different context lengths, highlighting the alignment between the inherent complexity of the queries and the models' capacity for holistic reasoning over extended contexts. However, hard queries consistently show the lowest performance across all context lengths, with a steep decline as context length extends, indicating the increasing challenge of reasoning over longer contexts with complex queries. Upon close inspection of failure cases, we observe that even though models can think step-by-step, they struggle to retrieve information for each condition for multi-conditional hard queries. Additionally, we find that when the volume of relevant information is too large, LCLMs complain about the input format being inappropriate and suggest using programmatic parsing. This points to an important research gap in the reasoning capability of LCLMs that demands further research. We observe similar trends for other models and provide further details in the Appendix A.5.

## 4.3 DISCUSSION

**Does the RAG help for holistic reasoning?** Holistic reasoning requires analyzing entire long contexts to answer questions comprehensively, making the RAG approach suboptimal for this task. RAG relies on retrieval models that focus on localized context retrieval, and they cannot effectively determine how much information should be retrieved. However, some studies suggest that reducing noise in the context may improve downstream tasks (Wu et al., 2024; Xu et al., 2024). With this in mind, we tested whether a retrieval model can filter irrelevant information and enhance holistic reasoning.

To test this hypothesis, we used constant information density and uniform information placement across models. We compared two large LCLMs, Llama-3.1-405b and GPT-4o, and a smaller LCLM, Llama-3.1-8b. As the document retriever, we used `BAAI/bge-large-en-v1.5`, a strong embedding-based retrieval model, retrieving 2k tokens with the highest similarity score to the query. In a shorter context (e.g., 4k tokens), the retrieval model might capture all relevant information, potentially enabling it to solve the holistic reasoning task without interference from extraneous data. However, in longer contexts, it becomes impossible to retrieve all relevant information, which tests the limits of this approach.

Figure 6 compares the performance of vanilla LCLMs and RAG-based models. For short contexts (4k tokens), RAG slightly outperforms vanilla LCLMs with GPT-4o and Llama-3.1-8b. This may be because, in this specific scenario, the retrieval model retrieves exactly the amount of relevant information, providing a cleaner context for holistic reasoning than the vanilla LCLM setting.

However, this configuration is not realistic for most practical scenarios, where we cannot know the exact amount of relevant information in advance. As the context length increases beyond 4k tokens, larger vanilla LCLMs consistently outperform RAG-based models. This suggests that LCLMs are more effective at handling longer contexts where RAG models may fail to retrieve sufficient relevant information.

Interestingly, with smaller models like Llama-3.1-8b, RAG shows better performance when the context length exceeds 16k tokens. This aligns with existing studies (Maekawa et al., 2024), which

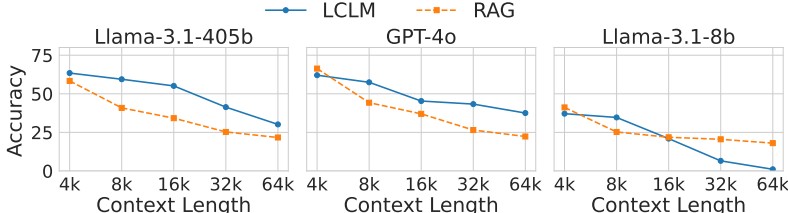

Figure 6: Performance comparison of LCLM and RAG. For long contexts, large models outperform RAG but a retriever helps a small model due to its limited ability to handle long contexts.

suggest that retrieval models can assist weaker models by mitigating their limitations of reasoning capability, even accounting for retrieval errors.

Further, detailed comparisons between RAG and LCLMs are provided in Appendix A.5.5. A promising future direction would be to develop a dynamic mechanism for determining the optimal amount of information to retrieve based on the query and context length, particularly when working with weaker models for holistic reasoning.

**Does CoT prompting improve performance?** We evaluated how chain-of-thought (CoT) prompting, which is our default configuration, impacts performance across different context lengths and query types (Wei et al., 2022; Kojima et al., 2022). We instructed LCLMs to provide answers directly without reasoning in a non-CoT setting. Figure 7 compares model performance with and without CoT prompting. The results indicate that CoT significantly and consistently enhances LCLM performance across all context lengths and query types. Specifically, aggregation queries are rarely solved without CoT, whereas max/min queries are relatively easier and less influenced by context length. These findings highlight the importance of CoT prompting in enabling LCLMs to handle complex holistic reasoning tasks more effectively, particularly in scenarios requiring deeper information synthesis.

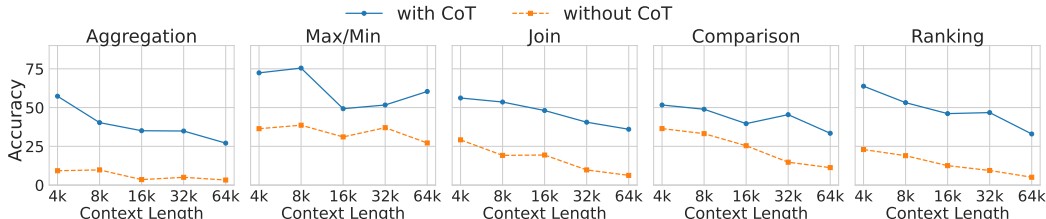

Figure 7: Performance comparison of GPT-4o with CoT and without CoT. Generally, CoT improves performance, particularly for tasks that require reasoning over multiple information sources.

## 5    CONCLUSION

In this paper, we introduced HoloBench, a novel benchmark designed to evaluate the holistic reasoning capabilities of LCLMs when handling complex, multi-document queries. Our framework systematically controls key factors such as context length, information density, information positioning, and query complexity, enabling a comprehensive assessment of LCLMs. Through extensive experiments, we demonstrated that the amount of information in the context has a larger impact on model performance than the actual context length. Additionally, certain query types, such as aggregation tasks, present unique challenges for LCLMs, especially as context length increases. This study fills a critical gap in evaluating LCLMs' holistic reasoning capabilities and provides valuable insights into their strengths and limitations in processing large-scale textual data. Future work could extend this framework to a broader range of reasoning tasks and explore optimization techniques to enhance LCLM performance in complex information aggregation scenarios.

## REPRODUCIBILITY STATEMENT

We have made several efforts to ensure the reproducibility of our work. First, the design principles, methodology, and evaluation setup for HoloBench are detailed in Sections 4 and 5. The SQL queries, natural language counterparts, and data preprocessing steps are provided in the Appendix (Sections A.2 and A.3.3). Moreover, the benchmark creation process, which includes database construction and verbalization steps, is clearly outlined in Section 4.2, and additional dataset details are available in the supplementary material.

To further enhance reproducibility, we will release all code and benchmark datasets, including queries, natural language questions, and corpora, upon paper acceptance. In all experiments, we set the temperature of LCLMs to 0.0, ensuring deterministic outputs for consistent model performance across multiple runs. All models used in the experiments are well-documented, with the prompt templates described in Appendix A.4. Upon paper acceptance, we will release the benchmark, codebase, and detailed documentation for both the experimental setup and data generation to promote ease of replication.

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

# A  APPENDIX

## A.1  MODEL DETAILS

We show the model details in Table 6.

| Model | Size | Context | HuggingFace / API |
|---|---|---|---|
| GPT-4o (OpenAI, 2024a) | - | 128k | gpt-4o-2024-08-06 |
| GPT-4o-mini (OpenAI, 2024a) | - | 128k | gpt-4o-mini-2024-07-18 |
| o3-mini (OpenAI, 2025) | - | 200k | o3-mini-2025-01-31 |
| o1-mini (OpenAI, 2024b) | - | 128k | o1-mini-2024-09-12 |
| Claude-3.5-Sonnet (Anthropic, 2024) | - | 200k | claude-3-5-sonnet@20240620 |
| Gemini-1.5-Pro (Reid et al., 2024) | - | 2M | gemini-1.5-pro-001 |
| Gemini-1.5-Flash (Reid et al., 2024) | - | 1M | gemini-1.5-flash-001 |
| Gemini-2.0-Flash (Reid et al., 2024) | - | 1M | gemini-2.0-flash-001 |
| Gemini-2.0-Flash-Thinking (Reid et al., 2024) | - | 1M | gemini-2.0-flash-thinking-exp-01-21 |
| Llama-3.1-405b (Dubey et al., 2024) | 405B | 128k | meta-llama/Llama-3.1-405B-Instruct |
| Llama-3.1-70b (Dubey et al., 2024) | 70B | 128k | meta-llama/Llama-3.1-70B-Instruct |
| Llama-3.1-8b (Dubey et al., 2024) | 8B | 128k | meta-llama/Llama-3.1-8B-Instruct |
| DeepSeek-V3 (DeepSeek-AI, 2024) | 671B | 128k | deepseek-ai/DeepSeek-V3 |
| DeepSeek-R1 (DeepSeek-AI, 2025) | 671B | 128k | deepseek-ai/DeepSeek-R1 |

Table 6: Models used in experiments. Model sizes are not publicly disclosed (-).

## A.2  DATASET DETAILS

### A.2.1  DATABASE PREPROCESS

To ensure that each verbalized document is comprehensive, we add entity name information for each table. This preprocessing involves mapping various identifiers (such as course IDs, instructor IDs, student IDs, etc.) to their corresponding human-readable names within the database. By integrating this descriptive information into the database tables, the data becomes more interpretable and contextualized, facilitating subsequent natural language processing tasks. Additionally, only entries with complete information are retained to ensure the consistency and reliability of the data, thereby improving the quality of the analysis. With this preprocessing, tables containing only identifiers and entity names are excluded from the verbalization process.

### A.2.2  DATASET SELECTION

To create HoloBench for evaluating LCLMs, we required large tables capable of generating long contexts. The following criteria were applied to ensure diversity in both domains and answers:

1. Databases must have more than two tables and at least 500 rows in total.

2. Each selected database must belong to a distinct domain.

Among the databases in the SPIDER dataset, only 15 meet the row count requirement:

```
['soccer_1', 'wta_1', 'baseball_1', 'sakila_1', 'flight_4',
'formula_1', 'college_2', 'bike_1', 'store_1', 'chinook_1',
'world_1', 'csu_1', 'flight_2', 'wine_1', 'car_1']
```

From these, we selected five databases spanning different domains while also diversifying the number of tables and rows. The selected databases are summarized in Table 7.

This selection balances domain variety, table count, and row diversity to create a robust evaluation benchmark.

| Base DB | # Tables | # Total Rows |
|---------|----------|--------------|
| Wine_1 | 3 | 555 |
| Store_1 | 4 | 2,548 |
| College_2 | 9 | 3,554 |
| Flight_4 | 3 | 20,989 |
| Soccer_1 | 5 | 185,608 |

Table 7: Summary of Selected Databases for HoloBench

### A.2.3 EXAMPLES OF DUMMY DOCUMENTS

In cases where a query pertains to all tables, resulting in an empty non-relevant table, we write ten dummy documents that do not affect any query result for each database. Example dummy documents are the following:

soccer_1: *Injuries are a common concern in soccer, affecting players' careers and team performance. Proper training, conditioning, and recovery protocols are essential to minimize risks. Teams invest in medical staff and technology to ensure players remain fit and healthy.*

wine_1: *Wine festivals celebrate the culture and craftsmanship of winemaking. These events often feature tastings, workshops, and opportunities to meet producers, allowing attendees to immerse themselves in the world of wine.*

## A.3 QUERY

### A.3.1 QUERY TYPE

The detailed explanation of each query type is described below:

1. **Aggregation:** Queries that perform data summarization using aggregate functions such as `SUM()`, `AVG()`, `COUNT()`, etc. These queries often involve combining multiple rows of data into a single summary value, usually with the use of `GROUP BY`.
   - *Example:* `SELECT SUM(sales) FROM transactions GROUP BY region`

2. **Max/Min:** Queries that retrieve the maximum or minimum value from a dataset. These queries use the `MAX()` or `MIN()` functions to find the highest or lowest value in a specified column.
   - *Example:* `SELECT MAX(salary) FROM employees`

3. **Join:** Queries that combine rows from two or more tables based on a related column between them. Joins such as `INNER JOIN`, `LEFT JOIN`, `RIGHT JOIN`, and `FULL JOIN` are used to retrieve data from multiple tables in a relational database.
   - *Example:* `SELECT employees.name, departments.name FROM employees INNER JOIN departments ON employees.department_id = departments.id`

4. **Comparison:** Queries that compare numerical values against a condition or threshold using comparison operators like `>`, `<`, `=`, `BETWEEN`, or `IN`. These queries filter records that meet specific criteria.
   - *Example:* `SELECT * FROM products WHERE price BETWEEN 100 AND 200`

5. **Ranking:** Queries that involve ordering results based on a certain metric, often using `ORDER BY` in conjunction with `LIMIT` to rank data and return only the top or bottom results.
   - *Example:* `SELECT player_name FROM players ORDER BY score DESC LIMIT 10`

### A.3.2 QUERY DIFFICULTY

The definition of our query difficulty is described below:

**Components Definition:**

- **SQL Components 1:** `WHERE, GROUP BY, ORDER BY, JOIN, OR, LIKE, HAVING, BETWEEN`

- **SQL Components 2:** EXCEPT, UNION, INTERSECT, NESTED subqueries (including EXISTS, IN, ANY, ALL with subqueries)
- **Other Complexity Factors:**
    - **Number of Aggregates:** More than 1 aggregate function (e.g., SUM, COUNT, AVG)
    - **Number of SELECT Columns:** More than 2 columns in the SELECT clause
    - **Number of WHERE Conditions:** More than 1 condition in the WHERE clause
    - **Number of JOIN Clauses:** More than 1 JOIN
    - **Number of GROUP BY Columns:** More than 1 column in the GROUP BY clause

**Difficulty Levels:**

1. **Easy:**
    - The query uses zero or exactly one keyword from **SQL Components 1**.
    - No keywords from **SQL Components 2**.
    - Does not satisfy any conditions under **Other Complexity Factors**.

2. **Medium:**
    - The query uses up to two keywords from **SQL Components 1**.
    - No keywords from **SQL Components 2**.
    - Satisfies no more than one condition from **Other Complexity Factors**.

3. **Hard:**
    - The query uses more than two keywords from **SQL Components 1**.
      **OR**
    - The query uses one or more keywords from **SQL Components 2**.
      **OR**
    - Satisfies more than one condition from **Other Complexity Factors**.

Since our questions are manually derived from SQL queries, more complex SQL queries with additional conditions result in longer and more intricate questions. To quantify this, we calculated the average word count per question across different levels of SQL query difficulty. The summarized results are presented in Table 8.

| SQL Difficulty | Average Question Length (words) |
|----------------|--------------------------------|
| Easy | 10.8 |
| Medium | 13.8 |
| Hard | 17.0 |

Table 8: Average Question Length by SQL Query Difficulty

### A.3.3 QUERY EXAMPLES

We give several example queries that are used in our experiments.

wine_1:

- **Question**: *Give me the average prices of wines that are produced by appellations in Sonoma County.*
- **Query**: SELECT AVG(T2.Price) FROM appellations AS T1 JOIN wine AS T2 ON T1.Appellation = T2.Appellation WHERE T1.County = 'Sonoma'
- **Type**: ['Aggregation', 'Join']
- **Difficulty**: Medium

store_1:

- **Question**: *How many tracks are longer than 12 minutes?*
- **Query**: `SELECT COUNT(DISTINCT name) FROM tracks WHERE milliseconds > 720000`
- **Type**: ['Aggregation', 'Comparison']
- **Difficulty**: Easy

`college_2`:

- **Question**: *Which course is taught in the building with the largest capacity classroom and what is the name of the building?*
- **Query**: `SELECT course.title, section.building FROM course JOIN section ON course.title = section.course_title JOIN classroom ON section.building = classroom.building WHERE classroom.capacity = (SELECT MAX(capacity) FROM classroom)`
- **Type**: ['Max/Min', 'Join']
- **Difficulty**: Hard

`flight_4`:

- **Question**: *Which are the top 3 airlines with the most active routes by counting the number of routes without codeshare for each airline?*
- **Query**: `SELECT airline_name, COUNT(*) AS active_routes FROM routes WHERE codeshare IS NULL GROUP BY airline_name ORDER BY active_routes DESC LIMIT 3`
- **Type**: ['Aggregation', 'Ranking']
- **Difficulty**: Hard

`soccer_1`:

- **Question**: *Which players are shorter than 170 cm?*
- **Query**: `SELECT player_name FROM Player WHERE height < 170`
- **Type**: ['Comparison']
- **Difficulty**: Easy

## A.4 PROMPT TEMPLATES

First, we provide a prompt used for our main experiments in Figure 8. Since o1-mini is a reasoning model, we remove the reasoning output part from the prompt for this model. Next, we show a prompt used for the "without CoT" setting in Figure 9. Also, Figure 10 shows an evaluation prompt.

## A.5 ADDITIONAL EXPERIMENTAL RESULTS

### A.5.1 INFORMATION POSITION

To generalize our insights described in Section 4.2.2, we additional conducted experiments for smaller models, GPT-4o-mini, Gemini-1.5 Flash, and Llama-3.1-8b, with all information positions and report results in Figure 11. We observe that smaller models also have their preferences of information positions. For example, GPT-4o-mini achieves better performance on "beginning" and "bimodal" positions than other positions. For Gemini-1.5 Flash, "end" and "bimodal" positions are better than other positions. Interestingly, Llama-3.1-8b achieves better performance not only "end" and "bimodal" but also "uniform" positions. Since this preference on "uniform" is shared with its larger model Llama-3.1-405b, their training process enhance models' information retrieval ability even if information pieces are distributed over documents.

You'll be given a set of sentences to read through carefully. Once you've reviewed them, I'll ask you a question related to the information in those sentences. Your job is to think critically about the details, analyze the sentences in relation to the question, and then provide your answer. If the information clearly supports a partial answer, provide that. However, if the evidence is unclear or insufficient, it is okay to respond with "No answer."

**Input:**
- **Sentences:**
" '
{context}
" '

- **Question:**
" '
{question}
" '

**Response:**
- **Reasoning:**
- [Describe how you thought through the sentences and how they helped you reach your conclusion. If the evidence is unclear or insufficient to provide a reliable answer, explain why. Your reasoning should not exceed 10,000 words.]
- **Answer:** [Provide an answer only if it is clearly supported by the information in the sentences. If the evidence is unclear or insufficient, respond with "No answer."]

Figure 8: Prompt template.

You'll be given a set of sentences to review carefully. Once you've reviewed them, I'll ask you a question related to the information in those sentences. Your task is to provide a direct, clear answer based only on what is clearly supported by the information. If the evidence does not fully support the answer, provide a partial answer. If the information is unclear or insufficient, respond with "No answer." Do not provide explanations or reasoning.

**Input:**
- **Sentences:**
" '
{context}
" '

- **Question:**
" '
{question}
" '

**Response:**
- **Answer:** [Provide only the direct final answer. If the information supports only part of the answer, provide a partial answer. If the evidence is unclear or insufficient, respond with "No answer." Do not include reasoning.]

Figure 9: Prompt template for the without CoT setting.

### A.5.2 QUERY TYPE

We demonstrate the results of o1-mini, Claude-3.5-Sonnet, Llama-3,1-70b, GPT-4o-mini, Gemini-1.5 Pro, Gemini-1.5 Flash, and Llama-3.1-8b in Figure 12. The experimental settings except models are the same as those in Figure 4, i.e., we use the proportional information amount and uniform information placement. We observe the same trend as Figure 12, i.e., all models achieve better performance on Max/Min queries than other queries and Aggregation queries are a harder task.

### A.5.3 QUERY DIFFICULTY LEVEL

We demonstrate the results of o1-mini, Claude-3.5-Sonnet, Llama-3,1-70b, GPT-4o-mini, Gemini-1.5 Pro, Gemini-1.5 Flash, and Llama-3.1-8b from the lens of query difficulty levels in Figure 13. We observe the same trend as Figure 5, the relative order of difficulty levels remains consistent over context lengths across various models.

### A.5.4 QUALITATIVE ANALYSIS

We observe that GPT-4o's behavior varies based on the context length provided, especially for aggregation queries, e.g., *"Find the destination airport and its country that receives flights from the highest number of distinct source airports, and include that number in the result."*. At 4k context length (Figure 14), GPT-4o delivers an exact match to the gold answer, correctly identifying "Kazan International Airport, Russia" with 7 distinct source airports. The concise context enables the model to effectively parse and enumerate the relevant information without error, providing an accurate response. As the context length increases to 8k (Figure 15), GPT-4o begins to struggle with correctly retrieving and aggregating the information. It returns the correct airport but miscounts the number of airports, representing a partial match. This is due to the limitation of models' retrieval ability over long context. This highlights a potential issue in the model's ability to maintain consistency in reasoning as context size increases, impacting its accuracy. When the context reaches 64k (Figure 16), the performance further deteriorates. GPT-4o produces a substantial error by incorrectly stating that Kazan International Airport has 10 distinct source airports, which fails to match the gold answer. Interestingly, unlike on 4k and 8k, it does not show the contents of its analysis over the sentence after describing a solution to the question.

### A.5.5 RAG COMPARISON

We conducted additional experiments to explore how different retrieval settings affect model performance.

First, we varied retrieval sizes and analyzed their impact on accuracy. We evaluated scenarios where half of a 64k context (32k tokens) contained query-relevant information and tested retrieval sizes of 2k, 4k, 8k, 16k, and 32k using the Llama-3.1-405b model. As shown in Table 9, performance peaked when the retrieval size (32k) matched the amount of relevant information, highlighting the benefit of minimizing irrelevant context to enhance reasoning.

| # Retrieved Tokens | 2k | 4k | 8k | 16k | 32k | 64k (LCLMs with full context) |
|---|---|---|---|---|---|---|
| RAG | 21.70 | 25.28 | 27.02 | 34.91 | 37.34 | 30.18 |

Table 9: Accuracy with Information Density = 0.5

In real-world scenarios, the density of relevant information is often unknown. To study this, we conducted experiments where all 64k tokens in the context were query-relevant. Note that absolute performance values are not directly comparable between scenarios with information density = 0.5 and 1.0 due to differing answer distributions. As shown in Table 10, using all 64k tokens yielded the highest performance, but the improvement over a 32k retrieval was marginal. This indicates that current LLMs struggle with ultra-long contexts, revealing significant opportunities to enhance long-context reasoning.

| # Retrieved Tokens | 2k | 4k | 8k | 16k | 32k | 64k (LCLMs with full context) |
|---|---|---|---|---|---|---|
| RAG | 14.85 | 19.18 | 21.79 | 29.36 | 38.69 | 39.06 |

Table 10: Accuracy with Information Density = 1.0

We examined whether RAG performs optimally when the distribution of relevant information is known. In a scenario with 32k relevant tokens within a 64k context, using the `bge-large-en-v1.5` retrieval system yielded the following metrics: **Precision**: 83.92%, **Recall**: 80.89%. These findings demonstrate that significant retrieval errors persist even under ideal conditions.

To summarize, RAG outperforms LCLMs when the retrieval size matches the relevant information. However, this alignment is rarely achievable in practical scenarios. Even with advanced retrieval models, retrieval errors remain significant , highlighting the difficulty of accurate information selection. Addressing long-context reasoning requires developing adaptive retrieval mechanisms that adjust dynamically to varying information densities or enhancing LCLMs' ability to process and reason over ultra-long contexts effectively. Simply increasing retrieval size or context length is insufficient and substantial improvements in both retrieval and model reasoning capabilities are essential for progress.

You will be given a question along with a response generated by an assistant and the corresponding ground truth data. Your task is to assess the response based on its accuracy and completeness in comparison to the ground truth. For each entry in the ground truth, determine whether the information provided by the assistant is an "Exact Match," a "Partial Match," or a "No Match."

#### **Evaluation Criteria:**
- **Exact Match**: The assistant's response precisely matches the ground truth in both content and detail.
- **Partial Match**: The assistant's response includes some correct information but is either incomplete, incorrectly ordered, or contains inaccuracies.
- **No Match**: The assistant's response does not accurately reflect the ground truth or is missing entirely.

#### **Special Cases:**
**Ground Truth is None**:
- If the ground truth is 'None' (represented as an empty list '[]'):
- **Exact Match**: If the assistant's response indicates that there is no information or content.
- **No Match**: If the assistant's response provides any information when the ground truth is 'None'.

#### **Output Format:**

- The output should be a list of objects where each object contains:
- An '"id"' that matches the 'id' of the corresponding ground truth entry.
- A '"label"' indicating whether the assistant's response is an '"Exact Match"', '"Partial Match"', or '"No Match"'.

- The number of output objects should match the number of entries in the ground truth.

—

### **Examples:**

{Several Examples}

====== Your task starts here ======

**Question:**
"'
{question}
"'

**Assistant's Response:**
"'
{pred}
"'

**Ground Truth:**
"'
{gold}
"'

**Output Format:**
"'
{output_format}
"'

Figure 10: Evaluation prompt.

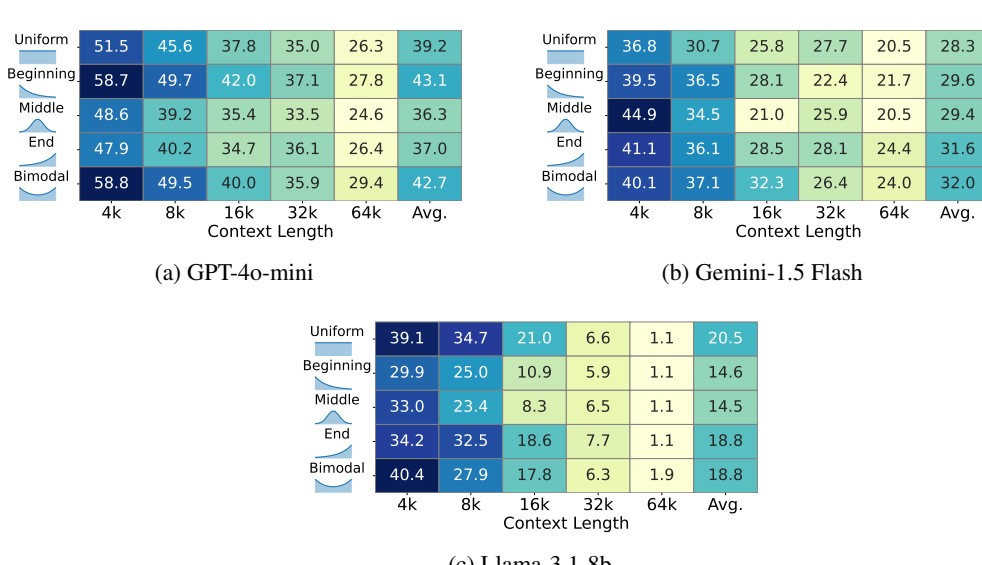

(a) GPT-4o-mini

(b) Gemini-1.5 Flash

(c) Llama-3.1-8b

Figure 11: Results of smaller models on various information positions.

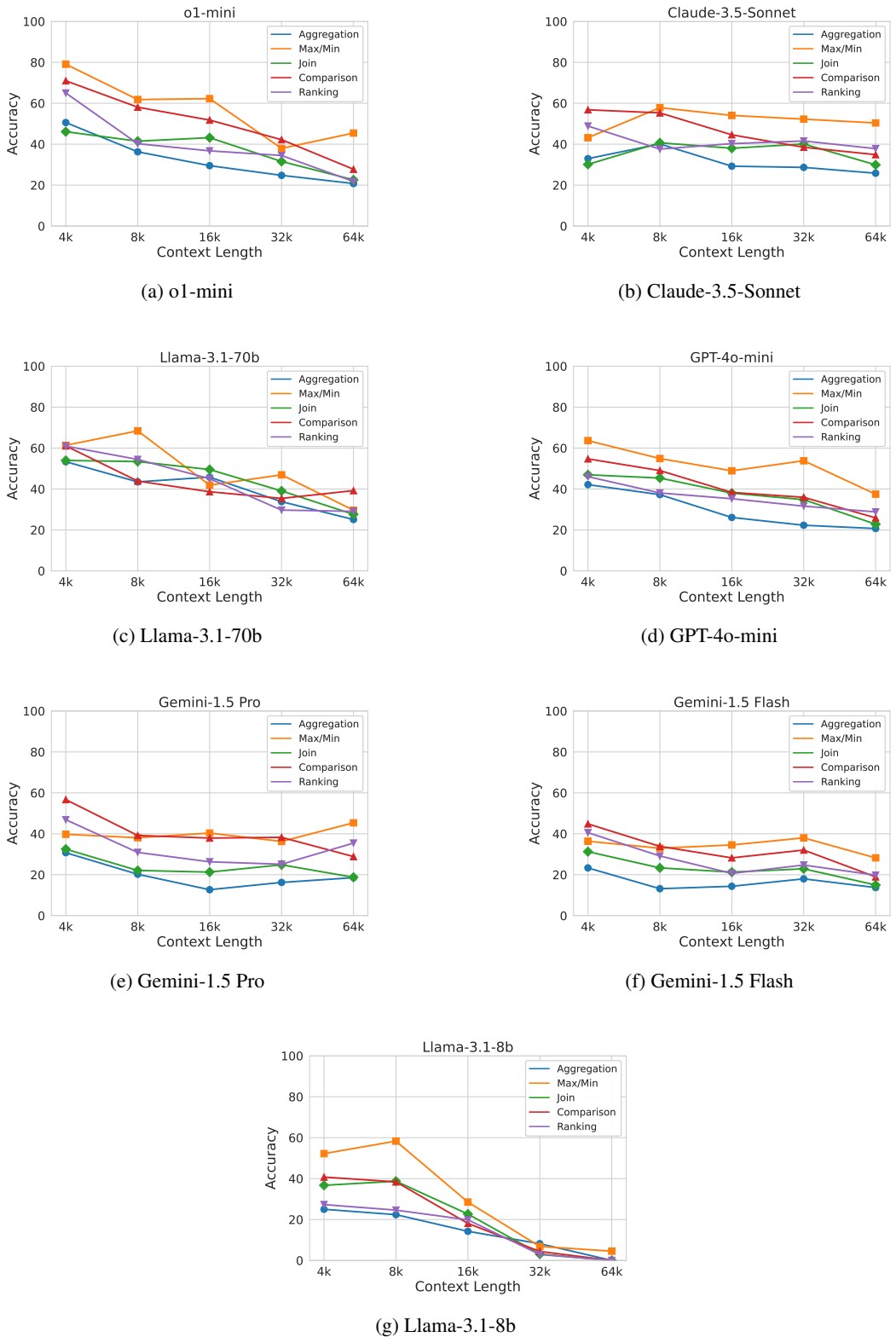

Figure 12: Performance of various models with proportional information amount across different query types.

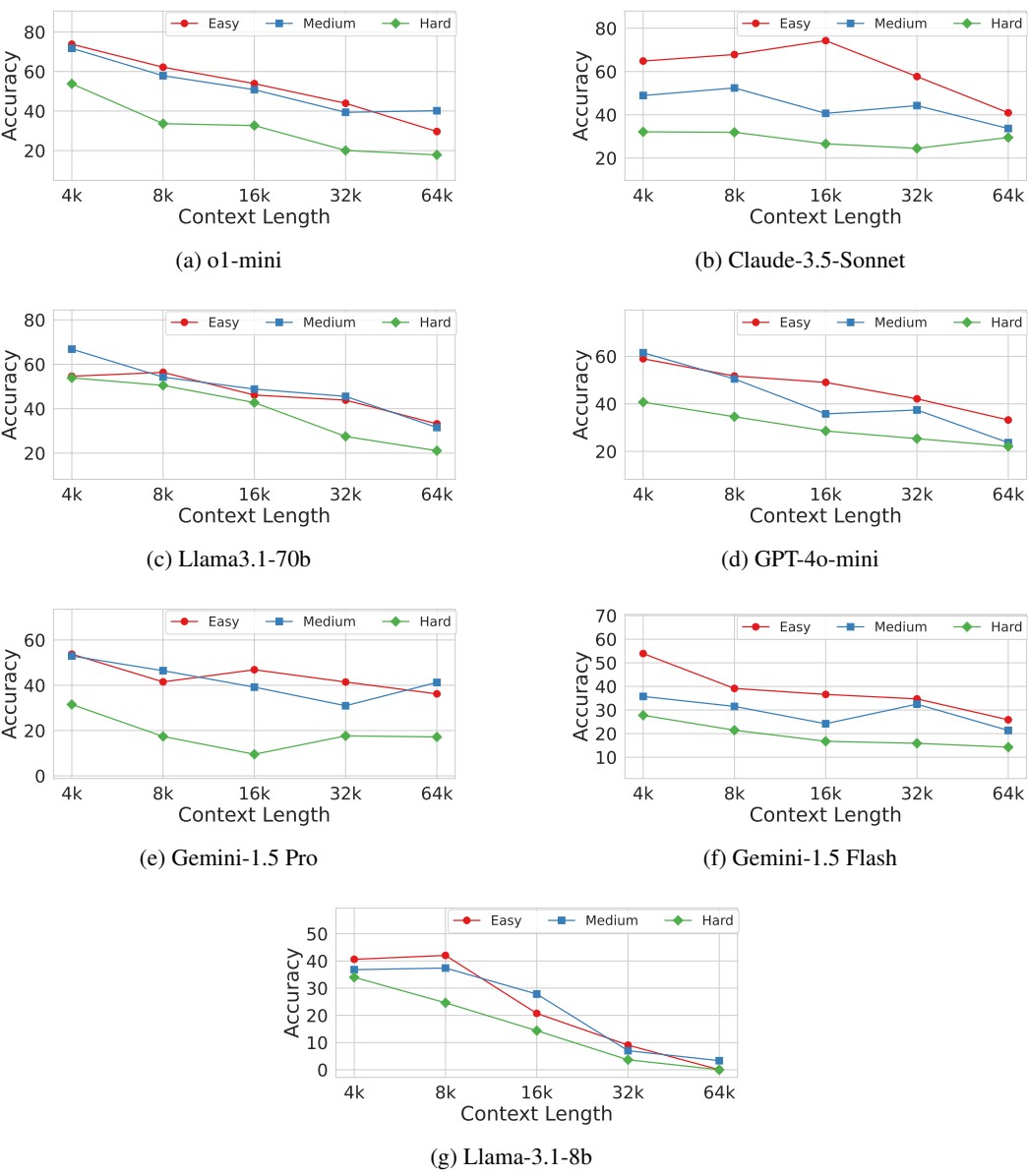

Figure 13: Performance of various models with proportional information amount across different difficulty levels.

> Question: *Find the destination airport and its country that receives flights from the highest number of distinct source airports, and include that number in the result.*
>
> ---
>
> **Reasoning:** ... I need to analyze the provided sentences and identify all the flights mentioned, ...
> 1. **Kazan International Airport, Russia:** - Source Airports: - Sochi International Airport ... - Koltsovo Airport - Number of distinct source airports: 7
> ... 10. **Yakutsk Airport, Russia:**... - Number of distinct source airports: 1
> After analyzing the data, Kazan International Airport in Russia receives flights from the highest number of distinct source airports, with a total of 7 distinct source airports.
> **Answer:** **Kazan International Airport, Russia; 7 distinct source airports. [Exact Match]**
>
> ---
>
> Gold Answer: "Kazan International Airport, "Russia", 7 distinct source airports

Figure 14: GPT-4o's output with 4k context. An evaluation score is shown after the model output and a gold answer is also provided below the dashed line. GPT-4o accurately list all relevant information and then provide a correct answer.

> - **Reasoning:** ... I reviewed the sentences for flight routes mentioned ... - The destination airport that appeared most frequently with different source airports was Kazan International Airport in Russia. It received flights from the following distinct source airports: - Sochi International Airport - Astrakhan Airport - Belgorod International Airport - Chelyabinsk Balandino Airport - Domodedovo International Airport - Pulkovo Airport ...
> - **Answer:** **Kazan International Airport, Russia; 6 distinct source airports. [Partial Match]**
>
> ---
>
> Gold Answer: "Kazan International Airport, "Russia", 7 distinct source airports

Figure 15: GPT-4o's output with 8k context. It retrieves most relevant information but misses one of relevant airports, resulting in Partial Match.

> **Reasoning:** ... I need to analyze the provided sentences and identify the destination airports along with their corresponding source airports. ...
> 1. **Identify Destination Airports:** I will go through the sentences to extract the destination ...
> 2. **Track Source Airports:** For each destination airport, I will keep track of the source airports ...
> ... 6. **Provide the Result:** The final result will include the destination airport, ...
> **Answer:** After analyzing the sentences, ... **Kazan International Airport** in **Russia**, with flights arriving from 10 distinct source airports. [No Match]
>
> ---
>
> Gold Answer: "Chengdu Shuangliu International Airport, "China", 40 distinct source airports

Figure 16: GPT-4o's output with 64k context. Instead of analyzing the context as it does with 4k and 8k contexts, it attempts to answer the question directly, resulting in an incorrect answer.

