# OpenReview forum: "Holistic Reasoning with Long-Context LMs: A Benchmark for Database Operations on Massive Textual Data"
_ICLR.cc/2025/Conference — ICLR 2025 Poster_

### Official Review · Reviewer_Epz7 · 2024-11-01

**Soundness:** 2
**Presentation:** 3
**Contribution:** 2
**Rating:** 6
**Confidence:** 4

**Summary:**

This paper argues that existing RAG systems struggle with complex tasks that require aggregation and reasoning over information spanning across multiple documents, while Long-Context language models may have the potential to this task but there is no study clearly show that. This paper proposed a new framework called HoloBench, which used database operations to generate context to test whether LCLMs can answer the target type of questions.
The experiments indicate that the amount of information in the context has a bigger influence on LCLM performance than the actual context length as well as the complexity of queries.

**Strengths:**

1. It is great to see a set of design principles in a paper that is to propose a benchmarking framework.

2. The paper did a comprehensive study of experiments and found interesting observations about factors, such as information amount, context length, information positions and query types and complexity to query performance.

**Weaknesses:**

1. The reality and the quality of the corpus/context is unclear. The author decided to generate the context by verbalizing data rows, which may lead to a set of unrealistic documents/passages with a simple description of attribute values in each row. Furthermore, there is no study in this paper that can show that the verbalization process is faithful to the original tables/rows.

2. The discussion about RAG and LCLMs are flawed. It does not make sense at all that limited RAG only retrieving 2k tokens and compared it with a language model that can be fed with 4k to 16k context.

**Questions:**

N/A

---

> ### Author Response · Authors · 2024-11-18
> **Author Response**
>
> Thank you for recognizing our comprehensive experimental analysis and clear design principles. We address your main concerns:
>
> 1. Context quality:
> Unlike model-based systems, we **ensure faithful and natural text generation** through manually crafted templates for each table (examples in Table 4). This template-based approach **guarantees complete preservation of the original tabular data** while maintaining fluent text, preventing hallucination or information loss that could occur with neural generation methods.
>
> 2. RAG Comparison:
> We acknowledge that our RAG setup in the original paper could have been more comprehensive. To address this concern, we conducted additional extensive experiments that provide deeper insights into the relationship between retrieval size and model performance.
>
> ## Additional Experiments with Varying Retrieval Sizes
>
> First, we investigated scenarios where half of the 64k context (i.e., 32k tokens) contains query-relevant information, examining performance trends across different retrieval sizes (2k, 4k, 8k, 16k, 32k). We used Llama-3.1-405b for these experiments.
>
> The results demonstrate that performance peaks when the retrieval size (32k) matches the actual amount of relevant information in the context. This supports our paper's argument that reducing unnecessary information can enhance reasoning capabilities.
>
> ### Table 1: Accuracy with information density = 0.5
>
> | # Retrieved Tokens | 2k | 4k | 8k | 16k | 32k | 64k (LCLMs with full context) |
> |-------------------|------|------|------|-------|-------|---------------------------|
> | RAG | 21.70 | 25.28 | 27.02 | 34.91 | 37.34 | 30.18 |
>
> ## Impact of Information Density
>
> However, in real-world applications, the density of relevant information isn't known a priori. To investigate this, we conducted experiments with contexts where all 64k tokens are query-relevant. Note that due to different answer distributions, the absolute values between information density = 0.5 and 1.0 scenarios aren't directly comparable.
>
> The results show that while using all 64k tokens achieved the best performance, the margin of improvement over 32k retrieval was surprisingly small. This suggests that current LCLMs still face limitations in processing ultra-long contexts, indicating significant room for improvement in their long-context reasoning capabilities.
>
> ### Table 2: Accuracy with information density = 1.0
>
> | # Retrieved Tokens | 2k | 4k | 8k | 16k | 32k | 64k (LCLMs with full context) |
> |-------------------|------|------|------|-------|-------|---------------------------|
> | RAG | 14.85 | 19.18 | 21.79 | 29.36 | 38.69 | 39.06 |
>
> ## Retrieval Quality Analysis
>
> We further investigated whether RAG would be optimal if we knew the distribution of relevant information beforehand. In scenarios with 32k relevant information within 64k context, our analysis using bge-large-en-v1.5 (a state-of-the-art retrieval system) achieved:
>
> - **Precision**: 83.92%
> - **Recall**: 80.89%
>
> These results, while strong, highlight that even in ideal settings, retrieval errors remain significant.
>
> ## Key findings:
>
> 1. While RAG can outperform LCLMs when the optimal retrieval size is known (matching the amount of relevant information), this information is rarely available in practical settings
> 2. Even with state-of-the-art retrieval models (bge-large-en-v1.5), substantial retrieval errors persist (Precision: 83.92%, Recall: 80.89%), highlighting the inherent challenges in information selection
> 3. The practical deployment of long-context reasoning systems requires either:
>    - Development of more adaptive and robust retrieval mechanisms that can dynamically adjust to varying information densities, or
>    - Enhancement of LCLMs' capabilities to process and reason over ultra-long contexts more effectively
>
> These findings suggest that the challenges in long-context reasoning cannot be solved by simply increasing retrieval size or context length, but rather require fundamental improvements in both retrieval quality and model capabilities.
>
> Thank you for your valuable feedback!

---

> > ### Author Response · Authors · 2024-11-27
> >
> > Dear Reviewer Epz7,
> >
> > Thank you again for reviewing our work! Please let us know if we have sufficiently addressed your questions and concerns. As the discussion deadline is approaching, we kindly request a prompt response and ask for a reconsideration of your evaluation score. Thank you for your support!
> >
> > Warm regards

---

> > ### Comment · Reviewer_Epz7 · 2024-11-28
> >
> > I appreciate the author's response to my reviews.
> > My concerns have been addressed, I will increase the score.

---

> > > ### Author Response · Authors · 2024-12-02
> > >
> > > Dear Reviewer Epz7,
> > >
> > > We are grateful for your positive response to our revisions and the updated evaluation. Your detailed feedback has been extremely valuable in refining our paper.

---

### Official Review · Reviewer_p5kH · 2024-11-04

**Soundness:** 2
**Presentation:** 3
**Contribution:** 2
**Rating:** 5
**Confidence:** 5

**Summary:**

The paper introduces HoloBench, a benchmark aimed at evaluating the holistic reasoning capabilities of Long-Context Language Models (LCLMs) when handling multi-document contexts. The framework aims to systematically assess LCLMs by controlling factors like context length, information density, distribution, and query complexity. The study uses a text-to-SQL dataset and corresponding databases to generate dataset for evaluating long-text language models. Based on proposed benchmark, the study shows that the quantity of information within the context more significantly impacts model performance than the pure length of the context. The findings provide insights into the strengths and limitations of LCLMs in processing large-scale text.

**Strengths:**

1.The paper offers a reasonable definition of the holistic reasoning capability of LLMs. The motivation of employing a text-to-SQL dataset to construct an evaluation framework for the holistic reasoning capabilities of Long-Context Language Models is reasonable.
2.The methodologies of benchmark construction are reasonable, for example, constructing data with varying difficulty levels and types based on the inherent difficulty classifications of the text-to-SQL dataset and the types of queries.
3.Given that the extractable information from databases can be freely controlled, the proposed Automated and Scalable Evaluation is practical.
4.The paper is written in a concise and coherent manner, offering illustrative figures that facilitate comprehension. The experiment results are clearly presented and easy to read.

**Weaknesses:**

1. Limited benchmark size:
According to Section 3.3, HoloBench consists of only 90 questions, which is significantly smaller in scale compared to other benchmarks like BABI Long and NOCHA. The small size of the benchmark makes it susceptible to randomness, compromising the reliability of the evaluation results.

2. Lack of diversity and practicality:
Since each database table stores data of a similar type, the data constructed by HoloBench tends to be highly homogeneous and confined to a single scenario dictated by the original database. Compared to other benchmarks that use richly varied books as a base corpus for long-context tasks, HoloBench, built from individual databases within a single text-to-SQL dataset, lacks diversity and richness, offering limited scenarios.

3. Limited challenge:
In HoloBench, rows from relevant and irrelevant tables are merged as relevant and irrelevant contexts, respectively. However, data stored in different tables of a database typically exhibit noticeable differences, allowing LLMs to potentially distinguish between relevant and irrelevant contexts with ease. This suggests that the tasks provided by HoloBench may not be sufficiently challenging.

4. Evaluation fairness and reliability:
HoloBench uses GPT-4-mini as its evaluation metric, which leads to potential uncontrollability and unfairness in evaluation. The human judgm

**Questions:**

Please answer the concerns in the above weaknesses section and the questions below:

1. Selection of databases:
According to reference [1], Spider consists of 200 different databases, yet as stated in line 214 of the paper, HoloBench only employs five databases from it. What was the reason behind selecting these specific five databases? Would the findings apply consistently across other databases and other scenarios as well?
[1] Spider: A large-scale human-labeled dataset for complex and cross-domain semantic parsing and text-to-SQL task


2. Potential information leakage:
The benchmark is constructed based on a publicly released text-to-SQL dataset from 2018, which might have already been included in the pre-training corpus of existing large language models. This could lead to potential information leakage, where LLMs might arrive at correct answers simply due to prior exposure to the data. Could the authors address this concern?

3. Comparative analysis using long-context models:
In Section 4.3, the authors utilize a document retriever that handles only 2k tokens while experimenting with data ranging from 4k to 64k tokens. This undermines the reliability of the performance comparison between LCLM and RAG. The authors should employ a RAG model that supports long-context processing to conduct this experiment accurately.

---

> ### Author Response · Authors · 2024-11-19
> **Author Response (1/2)**
>
> Thank you for your detailed review. We address each of your concerns:
>
> 1. Benchmark Size:
> Although HoloBench contains only 90 questions, each question is evaluated across numerous configurations, such as context length, information position, and information amount. For instance, testing 5 context lengths, 5 information positions, and 2 information amounts results in 4,500 test cases (90 × 5 × 5 × 2) per model—a scale unmatched by existing benchmarks mentioned. Moreover, we meticulously crafted our questions to encompass diverse query types and difficulty levels, with each configuration validated for correctness via SQL execution. This comprehensive coverage significantly amplifies the effective size of our evaluation, providing robust insights into model behavior across varying conditions, far beyond what the raw question count implies.
>
>
> 2 and 3. Diversity of databases and difficulty of benchmark:
> To create HoloBench for evaluating LCLMs, we required large tables capable of generating long contexts. Considering this, we applied the following criteria to ensure diversity in both domains and answers:
> Databases must have more than two tables and at least 500 rows in total.
> Each selected database must belong to a distinct domain.
> Among the databases in the SPIDER dataset, only 15 meet the row count requirement: ['soccer_1', 'wta_1', 'baseball_1', 'sakila_1', 'flight_4', 'formula_1', 'college_2', 'bike_1', 'store_1', 'chinook_1', 'world_1', 'csu_1', 'flight_2', 'wine_1', 'car_1’]. Then, we chose five databases in different domains.
> From these, we selected five databases spanning different domains while also diversifying the number of tables and rows. The selected databases are summarized below:
>
> | Base DB    | # Tables | # Total Rows |
> |------------|----------:|--------------:|
> | Wine_1     | 3        | 555          |
> | Store_1    | 4        | 2,548        |
> | College_2  | 9        | 3,554        |
> | Flight_4   | 3        | 20,989       |
> | Soccer_1   | 5        | 185,608      |
>
> This selection balances domain variety, table count, and row diversity to create a robust evaluation benchmark.
> Note that we did consider alternatives such as using external corpora (e.g., Wikipedia) to increase diversity and complexity in the context. However, ensuring control over the relevance of external information proved challenging, making it difficult to guarantee the correctness of the ground truth. To maintain a systematic and automated evaluation process, we chose to limit the context to the content within the databases.
> Although one might assume that LCLMs could easily distinguish between relevant and irrelevant contexts when they are drawn from different tables within a database, our results show that the models still face difficulties in making these distinctions.
>
> 3. Evaluation fairness and reliability:
> While we rely on GPT-4o-mini for automatic evaluation, we find the evaluations were aligned with human judgements (93.8% agreement).
>
> 4. Potential information leakage due to publicly released text-to-SQL dataset:
> While this is a valid concern, our benchmark is constructed by dynamically sampling table rows and generating contexts with relevant and irrelevant information. We derive the golden answers for different settings by executing SQL queries on the subtables. This makes it highly unlikely that LCLMs would have any parametric knowledge about the golden answers.

---

> ### Author Response · Authors · 2024-11-19
> **Author Response (2/2)**
>
> 5. RAG comparison:
> We acknowledge that our RAG setup in the original paper could have been more comprehensive. To address this concern, we conducted additional extensive experiments that provide deeper insights into the relationship between retrieval size and model performance.
>
> ## Additional Experiments with Varying Retrieval Sizes
>
> First, we investigated scenarios where half of the 64k context (i.e., 32k tokens) contains query-relevant information, examining performance trends across different retrieval sizes (2k, 4k, 8k, 16k, 32k). We used Llama-3.1-405b for these experiments.
>
> The results demonstrate that performance peaks when the retrieval size (32k) matches the actual amount of relevant information in the context. This supports our paper's argument that reducing unnecessary information can enhance reasoning capabilities.
>
> ### Table 1: Accuracy with information density = 0.5
>
> | # Retrieved Tokens | 2k | 4k | 8k | 16k | 32k | 64k (LCLMs with full context) |
> |-------------------|------|------|------|-------|-------|---------------------------|
> | RAG | 21.70 | 25.28 | 27.02 | 34.91 | 37.34 | 30.18 |
>
> ## Impact of Information Density
>
> However, in real-world applications, the density of relevant information isn't known a priori. To investigate this, we conducted experiments with contexts where all 64k tokens are query-relevant. Note that due to different answer distributions, the absolute values between information density = 0.5 and 1.0 scenarios aren't directly comparable.
>
> The results show that while using all 64k tokens achieved the best performance, the margin of improvement over 32k retrieval was surprisingly small. This suggests that current LCLMs still face limitations in processing ultra-long contexts, indicating significant room for improvement in their long-context reasoning capabilities.
>
> ### Table 2: Accuracy with information density = 1.0
>
> | # Retrieved Tokens | 2k | 4k | 8k | 16k | 32k | 64k (LCLMs with full context) |
> |-------------------|------|------|------|-------|-------|---------------------------|
> | RAG | 14.85 | 19.18 | 21.79 | 29.36 | 38.69 | 39.06 |
>
> ## Retrieval Quality Analysis
>
> We further investigated whether RAG would be optimal if we knew the distribution of relevant information beforehand. In scenarios with 32k relevant information within 64k context, our analysis using bge-large-en-v1.5 (a state-of-the-art retrieval system) achieved:
>
> - **Precision**: 83.92%
> - **Recall**: 80.89%
>
> These results, while strong, highlight that even in ideal settings, retrieval errors remain significant.
>
> ## Key findings:
>
> 1. While RAG can outperform LCLMs when the optimal retrieval size is known (matching the amount of relevant information), this information is rarely available in practical settings
> 2. Even with state-of-the-art retrieval models (bge-large-en-v1.5), substantial retrieval errors persist (Precision: 83.92%, Recall: 80.89%), highlighting the inherent challenges in information selection
> 3. The practical deployment of long-context reasoning systems requires either:
>    - Development of more adaptive and robust retrieval mechanisms that can dynamically adjust to varying information densities, or
>    - Enhancement of LCLMs' capabilities to process and reason over ultra-long contexts more effectively
>
> These findings suggest that the challenges in long-context reasoning cannot be solved by simply increasing retrieval size or context length, but rather require fundamental improvements in both retrieval quality and model capabilities.
>
> Thank you for your valuable feedback!

---

> > ### Author Response · Authors · 2024-11-27
> >
> > Dear Reviewer p5kH,
> >
> > Thank you again for reviewing our work! Please let us know if we have sufficiently addressed your questions and concerns. As the discussion deadline is approaching, we kindly request a prompt response and ask for a reconsideration of your evaluation score. Thank you for your support!
> >
> > Warm regards

---

> > > ### Author Response · Authors · 2024-12-03
> > >
> > > Dear Reviewer p5kH,
> > >
> > > Thank you again for your feedback and time. As today is the final day for score updates, we kindly ask if our response and revisions have sufficiently addressed your concerns and if you might reconsider your evaluation score.
> > >
> > > We greatly appreciate your support!

---

### Official Review · Reviewer_53mu · 2024-11-04

**Soundness:** 4
**Presentation:** 4
**Contribution:** 4
**Rating:** 8
**Confidence:** 4

**Summary:**

The authors propose a new comprehensive benchmark to test the reasoning capabilities of LCLM across multiple dimensions. The authors unveil a range of novel insights from this benchmark, by comparing the performance of recent LCLM on diverse scenario (position of the information, density of information, complexity of the query).

**Strengths:**

The authors :
- Proposed a new, original, comprehensive and timely benchmark dataset and framework based on text-to-sql for consistent long context LM benchmarking
- Important benchmark dataset and frameworks, showing novelty into aspects of LCLM seemingly not thoroughly explored yet
- Identified a range of novel and important insights about LCLM reasoning (lost in the middle comparison, model size, RAG comparison, query complexity effect, CoT importance)
- Tested recent LCLM (Llama 3.1, GPT4o, Claude 3.5, Gemini 1.5)
- Used LLM-based evaluation metrics, but which showed 93.8% agreement with human judgements
- Good paper layout to quickly identify important insights derived from the benchmark
- Good discussion points on RAG contribution

**Weaknesses:**

- Minor: "Information density" concept could be further detailed

**Questions:**

/

---

### Official Review · Reviewer_KK9H · 2024-11-04

**Soundness:** 3
**Presentation:** 4
**Contribution:** 3
**Rating:** 6
**Confidence:** 3

**Summary:**

This paper presents HoloBench, a benchmark designed to evaluate the holistic reasoning capabilities of Long-Context Language Models (LCLMs). While Retrieval-Augmented Generators (RAGs) effectively access information from large document collections, they struggle with complex tasks that require aggregation and reasoning across multiple documents. The authors refer to this as holistic reasoning, for which long-context language models have been developed. HoloBench aims to evaluate LCLMs against RAGs under various conditions.

Although similar datasets have been created in the past, this dataset is more comprehensive in terms of context length, information density, information distribution, and query complexity, and it includes automated evaluation methods.

**Strengths:**

The code has been released, and the data is reproducible. I believe the authors are providing a valuable service to the community.

**Weaknesses:**

From a technical standpoint, this method is relatively straightforward.

**Questions:**

How do you differentiate between long but simple questions and genuinely complex ones?

---

> ### Author Response · Authors · 2024-11-18
> **Author Response**
>
> Thank you for your positive feedback about our benchmark's potential value to the community and the reproducibility of our work.
>
> > How to differentiate b/w long but simple questions and genuinely complex ones
>
> Since we manually create questions based on SQL queries, the more complex the SQL query, the more conditions it contains, leading to longer questions. So, longer questions generally tend to be more complex questions.
>
> To quantify this, we calculated the average number of words per question for each level of SQL query difficulty. The results are as follows:
>
> | SQL Difficulty | Average Question Length (words) |
> |:--------------:|--------------------------------:|
> | Easy          | 10.8                            |
> | Medium        | 13.8                            |
> | Hard          | 17.0                            |
>
> This confirms a trend within our dataset: questions derived from more complex queries tend to be longer.
>
> If we have misunderstood your point, please let us know.

---

> > ### Author Response · Authors · 2024-11-27
> >
> > Dear Reviewer KK9H,
> >
> > Thank you again for reviewing our work! Please let us know if we have sufficiently addressed your questions and concerns. As the discussion deadline is approaching, we kindly request a prompt response and ask for a reconsideration of your evaluation score. Thank you for your support!
> >
> > Warm regards

---

### Author Response · Authors · 2024-11-26
**General Response**

We sincerely appreciate your time and constructive feedback, which have greatly enhanced the quality of our work. In response to your suggestions, we have uploaded a revised version of our paper, with all the corrections highlighted in blue for your convenience. Below is a summary of the key revisions:
- Appendix A.1.2: We have detailed our methodology for selecting diverse databases and provided comprehensive statistics for each.
- Appendix A.2.2: We included a discussion on query length to quantify how longer questions tend to be more complex.
- Appendix A.4.5: We offered deeper insights into the relationship between RAG and LCLMs.

As the rebuttal phase concludes, we would greatly appreciate it if you could update your review based on these revisions.

---

### Meta-Review · Area_Chair_yqgH · 2024-12-21

**Metareview:**

In this paper, the authors propose a benchmark for evaluating the capabilities of long-context language models. Reviewers consider this a valuable contribution to the community, given the release of the associated code and data, which facilitate comparisons between long-context LMs and RAG approaches. They also found the paper well-written and noted meaningful insights from the experiments. While there were discussions regarding the significance of the work, the authors provided a high quality response. Overall, I believe this paper exceeds the acceptance threshold.

**Additional Comments On Reviewer Discussion:**

The authors effectively addressed concerns about the significance of the work, as well as the scale, difficulty, and diversity of the benchmark. They also presented additional experiments to compare long-context LMs with RAG approaches. Some reviewers responded positively and raised their scores after reviewing these updates.

---

### Decision · Program_Chairs · 2025-01-22

Accept (Poster)